# Clinical Progression of Metabolic-Associated Fatty Liver Disease Is Rare in a Danish Tertiary Liver Center

**DOI:** 10.3390/jcm11092271

**Published:** 2022-04-19

**Authors:** Tea Lund Laursen, Mikkel Breinholt Kjær, Louise Kristensen, Henning Grønbæk

**Affiliations:** Department of Hepatology and Gastroenterology, Aarhus University Hospital, 8200 Aarhus, Denmark; tealaurs@rm.dk (T.L.L.); mikj@clin.au.dk (M.B.K.); louisek92@hotmail.com (L.K.)

**Keywords:** MAFLD, fibrosis, inflammation, steatosis, liver biopsy

## Abstract

Data concerning non-invasive discrimination of simple steatosis from steatohepatitis in metabolic-associated fatty liver disease (MAFLD) and risk of disease progression in patients with MAFLD are conflicting. We aimed to investigate these factors in an MAFLD cohort at a Danish tertiary liver centre. We retrospectively assessed 129 patients with biopsy-proven MAFLD. Patients were divided according to the presence of simple steatosis or steatohepatitis in liver biopsies. Histological and clinical progression were assessed during follow-up. Patients with steatohepatitis had higher BMIs, liver stiffness, HbA1c and soluble (sCD163) and were more prone to have metabolic syndrome at baseline compared with simple steatosis patients. Of the 129 patients, 31 had a follow-up biopsy after a median of 287 days; simple steatosis progressed to steatohepatitis in 7 cases, while 2 regressed. Twenty patients had the same fibrosis stage according to the follow-up biopsy, seven progressed and four regressed. Only 14 patients progressed clinically (median follow-up time was 3.8 years). Clinical progression was associated with female sex, high creatinine, high sCD163 and disease severity in the diagnostic liver biopsy. Steatohepatitis was associated with metabolic and inflammatory parameters including fibroscan. Disease progression was seen in only 11% of cases and was mainly related to more severe histological disease at baseline.

## 1. Introduction

An international consensus paper has suggested changing the nomenclature of non-alcoholic fatty liver disease (NAFLD) to the new term, metabolic-associated fatty liver disease (MAFLD), focusing on positive findings rather than excluding other etiologies [1,2]. We will adhere to this terminology in the present paper. The alarming increase in the prevalence of obesity and metabolic dysfunction has contributed to a dramatic increase in the prevalence MAFLD, which is now considered to affect approximately 25% of the world population [3,4]. Most often, MAFLD is asymptomatic and presents with simple steatosis on liver biopsies, but this may progress to inflammation, i.e., steatohepatitis and potentially fibrosis and cirrhosis associated with clinical decompensation, liver transplantation or early death [5]. MAFLD is closely associated with obesity and type 2 diabetes mellitus (T2DM) and is considered the hepatic manifestation of the metabolic syndrome (MS) [4,6].

A number of studies have aimed to describe characteristics capable of predicting the risk of progression and the severity of disease progression. Despite these efforts, controversy remains with some studies indicating a high risk of progression [7], whereas others indicate a much lower risk [8,9]; however, up to 20% may be rapid progressors [10].

Thus, the aim of the present study was to evaluate a cohort of patients with biopsy-verified MAFLD at a tertiary liver center in Denmark, to asses which factors separate simple steatosis from steatohepatitis at diagnosis and to investigate how many patients will experience disease progression and if any baseline factors may predict this progression. Furthermore, we facilitated the newly proposed diagnostic criteria for MAFLD and evaluated the accuracy of these compared to histological NAFLD diagnosed by liver biopsy.

## 2. Materials and Methods

### 2.1. Design and Patients

We retrospectively identified patients who underwent a liver biopsy at the Department of Hepatology and Gastroenterology, Aarhus University Hospital between 2008 and 2018 with a diagnosis of NAFLD based on ICD-10 diagnostic codes (DK760, DK760A, DK760B, DK760C, DR740). The patients were identified using a combination of the diagnostic codes and the following procedure codes: liver biopsy (KJJA20) or transjugular biopsy (KJJA26). Patients were included if the diagnostic biopsy showed steatosis interpreted as NAFLD by the pathologist. The patients were excluded if the biopsy or patient history revealed signs of other liver diseases or causes, e.g., autoimmune hepatitis, hepatitis B (HBV), hepatitis C (HCV), alcohol abuse or drug-induced liver injury (DILI). Figure 1 summarizes study recruitment as a flow diagram. The MAFLD diagnosis was based on the positive diagnostic criteria for MAFLD proposed in a recent consensus statement [1,2].

All patients had evidence of steatosis in the liver biopsy. Patients with overweight/obesity (defined by a BMI ≥ 25 kg/m^2^) or T2DM defined by (HbA1c > 48 mmol/L or on antidiabetic medicine) were classified as MAFLD. Further, lean or normal-weight patients (defined as BMI < 25) who fulfilled at least two of the following metabolic risk abnormalities were classified as MAFLD: (1) waist circumference ≥102 cm or 88 cm in men and women, respectively, (2) hypertension (>130/85 mmHg or antihypertensive treatment), (3) plasma triglycerides (TGs) ≥ 1.70 mmol/L, (4) plasma HDL cholesterol <1.00 mmol/L or <1.29 mmol/L in men and women, respectively, (5) prediabetes (HbA1c 39–47 mmol/L), (6) homeostasis model assessment ≥2.5 and (7) plasma high-sensitivity C-reactive protein level (>2 mg/L).

The study was approved by the Danish Data Protection Agency (1-16-02-693-18) and the Danish Health Authorities (3-3013-2948/1). The date of diagnosis was defined as the day the patient received the diagnosis. Initial treatment was started when the patient was diagnosed or immediately after it was decided whether the patient should participate in a clinical trial.

### 2.2. Liver Biopsies

All liver biopsies were evaluated and scored by an experienced hepato-pathologist. Steatosis was scored 1–3 (grade 1: 5–33% of the surface area involved by steatosis as evaluated on low-to-medium-power examination, grade 2: 33–66%, grade 3: >66% steatosis), ballooning and inflammation 0–2 and fibrosis 0–4, including 1a-c. Patients were divided into groups of simple steatosis or steatohepatitis based on the steatosis, activity and fibrosis (SAF) score [11] or the NAFLD activity score (NAS) [12] (<5 vs. ≥5). If more than two biopsies were available, the first and the last were used as the diagnostic and the follow-up biopsy.

### 2.3. Study Parameters

Data concerning lifestyle, risk factors, ultrasound findings, transient elastography, comorbidities and biochemical variables were extracted from the patients’ electronic charts and entered into the REDCap database. Comorbidities were considered present if the patient received medication or was diagnosed by a doctor. Metabolic syndrome (MS) was defined according to the International Diabetes Federation [13] as the presence of obesity (abdominal circumference >94 cm for men and >80 cm for women or BMI > 30) and two or more of the following: lipid abnormalities (triglycerides > 1.7 mmol/L or hypercholesterolemia), low levels of high-density lipoprotein (HDL) (<1.03 mmol/L for men and <1.29 mmol/L for women), hypertension and T2DM.

### 2.4. Disease Progression and Regression

Disease progression was separated into histological or clinical progression. Histological progression was defined as simple steatosis on the diagnostic biopsy with steatohepatitis on the follow-up biopsy; regression was considered to be the opposite, and in addition, whether fibrosis progressed, regressed or remained stable as defined by fibrosis stages. Clinical progression was defined as the occurrence of cardiovascular events, newly diagnosed diabetes, cirrhosis, hepatocellular carcinoma (HCC), liver decompensation (varices, hepatic encephalopathy or ascites) or liver-related death between the diagnostic biopsy and end of follow-up. Patients were defined as lost to follow-up if no other data than that gathered from the day of diagnosis were available in the electronic patient chart.

### 2.5. Statistical Analysis

Between-group comparisons were performed using Student’s t-test on (log)normally distributed data or the Wilcoxon rank sum test for non-normally distributed data. Comparisons of binary data were performed with the chi-square test or Fisher’s exact test according to sample size. Logistic regression analyses were performed to assess if variables were independently associated with histological or clinical progression. Data are expressed as medians with interquartile ranges and as numbers with proportions unless otherwise specified. A *p*-value below 0.05 was considered statistically significant. The data were analyzed in Stata 16 (StataCorp, College Station, TX, USA).

## 3. Results

### 3.1. Patients and Follow-Up

In total, 260 consecutive patients were identified, and 135 patients with a diagnosis of NAFLD were included. By facilitating the novel diagnosis of MAFLD 129/135 (96%), patients met the criteria. The median age of MAFLD patients at diagnosis was 46 (31–56) years, and the median BMI was 33 kg/m^2^ (29–36).

The six patients that did not meet MAFLD diagnostic criteria differed from MAFLD patients by having a lower BMI (23 (IQR 20–23) kg/m^2^) and increased levels of bilirubin (14 (IQR 10–14) µmol/L). The baseline characteristics of the 129 included patients are listed in Table 1.

Based on liver biopsies, MAFLD patients were categorized as having simple steatosis or steatohepatitis. According to the SAF score, 72 (56%) patients had simple steatosis and 57 (44%) had MAFLD with steatohepatitis. According to the NAS score, 80 (62%) patients had steatosis whilst 49 (38%) had steatohepatitis. In the diagnostic biopsy, 63 patients had F0 fibrosis, 44 had F1 fibrosis, 13 had F2 fibrosis, 6 had F3 fibrosis and 3 had cirrhosis (F4 fibrosis). Median follow-up time was 3.8 years, but with a wide range (12 days–27 years). During follow-up, 31 patients had an additional biopsy performed, and the median time between biopsies was 287 days, again with a wide range (172 days–25.5 years). Sixty patients were initially treated with lifestyle intervention, and significantly more patients were treated with lifestyle interventions in the MAFLD with steatohepatitis group compared to simple steatosis groups. During follow-up, 25 subjects initiated new treatment for MAFLD or T2D; A total of 16 of these initiated vitamin E therapy.

### 3.2. Parameters to Separate Simple Steatosis from MAFLD with Steatohepatitis at Baseline

The patients with MAFLD with steatohepatitis (based on SAF ≥ 5) had a significantly higher BMI, liver stiffness, fibrosis grade and activity, and more patients had MS compared to patients with simple steatosis (Table 1). In addition, steatohepatitis patients had significantly higher sCD163, HbA1c, triglycerides and CRP compared to patients with simple steatosis, while amylase, HDL and LDL were lower. Gamma-glutamyltransferase (GGT) tended to be elevated in patients with steatohepatitis (*p* = 0.053). For multiple regression analyses including all the single predictors, sCD163 was the only independent predictor of steatohepatitis for both SAF and NAS scoring systems. There were no significant differences in age, sex, alcohol consumption or smoking habits between the groups.

### 3.3. Histological Progression

Of the 31 MAFLD patients with a follow-up biopsy, 25 had simple steatosis and 6 had steatohepatitis according to the diagnostic biopsy when using SAF score. According to NAS scores < 5 and ≥5, 28 had simple steatosis and 3 had steatohepatitis, respectively. Four patients with follow-up biopsies were initially treated with vitamin E, which had no impact on follow-up histology. According to the SAF score, simple steatosis progressed to steatohepatitis in 7 of 25 (28%) cases, while 2 of 6 (33%) patients regressed from steatohepatitis to simple steatosis. When using the NAS score, 8 of 28 (28%) patients progressed and 1 of 3 regressed (33%). Twenty (64%) patients had the same fibrosis stage for the follow-up-biopsy, seven (23%) progressed, and four (13%) regressed. There was no significant difference in fibrosis stage between baseline and follow-up biopsies and no difference in the risk of fibrosis progression between MAFLD with simple steatosis and MAFLD with steatohepatitis groups. Out of the 25 patients with MAFLD with simple steatosis, according to SAF scoring, 6 (24%) had progression of fibrosis, 3 (12%) regressed, and 16 (64%) had the same fibrosis stage for the follow-up biopsy.

### 3.4. Clinical Progression

Median time to clinical progression was 25 months (range: 5–112 months). Of the 129 MAFLD patients, 14 (11%) progressed, 88 (68%) did not, and 27 (21%) were categorized as lost to follow-up. Only one (0.8%) MAFLD patient progressed to cirrhosis, and none developed decompensated liver disease or HCC. Two patients had cardiovascular events; one thrombus and one with several deep vein thromboses but also a factor V Leiden mutation. One cirrhosis patient developed portal hypertensive gastropathy. Ten patients (8%) were diagnosed with T2DM during follow-up. Only one person died; this was not liver-related. During follow-up, 25 patients started a new treatment, of which 16 started vitamin E supplementation, 7 patients started GLP-1 analogue treatment for T2DM, and 2 patients started metformin treatment. Treatment with vitamin E had no influence on clinical progression. Risk factors for clinical progression were: female sex, low creatinine, high sCD163, high NAS score, high activity score and high ballooning score on the diagnostic biopsy. A high fibrosis stage in the diagnostic biopsy tended to be associated with clinical progression (*p* = 0.07) (Table 2). In multiple regression analysis, there were no independent predictors for disease progression.

## 4. Discussion

The primary aim of this study was to assess the natural history of fatty liver disease in a cohort of Danish patients with biopsy-verified NAFLD and investigate clinical, biochemical and histological factors associated with disease progression. Furthermore, we adhered to the new definition of MAFLD.

At baseline, individuals with MAFLD differed from individuals who solely fulfilled the criteria for a diagnosis of NAFLD. Non-MAFLD NAFLD patients were leaner, less likely to have MS and exhibited near-normal metabolic biochemical parameters. Although the small number of individuals in this group (*n* = 6) makes meaningful interpretation difficult, these results mirror data from recent publications [14,15]. Nguyen et al. showed that patients fulfilling NAFLD but not MAFLD criteria had a lower prevalence of overweight, obesity, prediabetes, diabetes and lower triglycerides compared to patients diagnosed with MAFLD compared to NAFLD [14].

In our MAFLD group, individuals with steatohepatitis differed clinically from the individuals with simple steatosis by having a significantly higher BMI and increased liver stiffness, and a larger proportion had T2DM at baseline. Furthermore, individuals with steatohepatitis presented with deranged biochemistry, namely significantly higher levels of TGs, HbA1c, and sCD163 and significantly lower levels of HDL, LDL and amylase compared to individuals with MAFLD with simple steatosis. These findings suggest a more severe metabolic dysfunction in individuals with MAFLD with steatohepatitis compared to simple steatosis. During follow-up, a further 10 individuals in the MAFLD group developed T2DM. The importance of this is underlined by studies showing increased mortality with increasing severity of metabolic dysregulation [16,17]. Stephanova et al. found that the presence of T2DM was an independent predictor of overall mortality in NAFLD patients while insulin resistance, T2DM, obesity and MS were independently associated with liver-related mortality [17]. Similarly, Golabi et al. found that the proportion of overall deaths was 34 times higher in NAFLD patients with a minimum of one component of MS compared to NAFLD patients with no components of MS. Furthermore, with every additional component of MS, the magnitude of risk of death increased [16]. This illustrates that specific care should be aimed at MAFLD patients with several metabolic risk factors.

Among the 129 patients with a diagnosis of MAFLD, 31 had liver biopsy performed during follow-up. Progression from simple steatosis to steatohepatitis based on SAF score was seen in 28% of patients during the median follow-up of 287 days while two of six (33%) patients with MAFLD with steatohepatitis regressed.

MAFLD displays large interindividual heterogeneity and great temporal dynamics, which complicates accurate assessment of the risk of progression. Despite this, MAFLD with steatohepatitis is believed to be a more severe subtype than MAFLD with simple steatosis, exhibiting higher mortality [18] and an increased fibrosis progression rate [10]. As fibrosis stage is the most important predictor of mortality in MAFLD [3,10], knowledge on the natural history of fibrosis development in these patients is prognostically important.

In the follow-up biopsy, seven individuals (23%) had progression of fibrosis, while four (13%) had regression. Only one patient progressed to cirrhosis. Interestingly, no statistical difference in the risk of fibrosis progression was found between individuals with MAFLD with steatohepatitis and individuals with MAFLD with simple steatosis. This finding places our study along recent studies challenging the dogma that MAFLD with simple steatosis is a benign condition that rarely progresses to steatohepatitis and fibrotic disease [19,20,21]. Furthermore, the dynamic nature of disease found in this study is in line with previous studies showing frequent progression and regression of both disease activity and fibrosis [10,18,22].

Clinical progression in this study was rare with a total of 11% of patients progressing and 68% exhibiting stable disease; however, 21% were lost to follow-up. Median time to clinical progression was 25 months (range: 5–112 months). Only 2 of the 14 individuals who progressed clinically developed liver-related complications; A total of 2 experienced cardiovascular events while for the majority of patients, clinical progression was due to a diagnosis of T2DM. Although MAFLD is an increasingly prevalent cause of cirrhosis and hepatocellular carcinoma (HCC) [23], cardiovascular disease continues to be the leading cause of death in MAFLD [24]. Furthermore, the large incidence of T2DM in this cohort underlines the bidirectional association between MAFLD and metabolic dysregulation. Risk factors for clinical progression included female sex, low creatinine, high sCD163 and more severe disease in the diagnostic biopsy; however, we could not define any independent predictors for disease progression.

In our study, sCD163 was the only independent predictor of MAFLD with steatohepatitis, and recent studies showed that this marker of macrophage activation is associated with morphological disease grade and that a high concentration of sCD163 predicts advanced fibrosis (>F3 or above) in NAFLD [25,26]. Recent studies demonstrated a significant correlation between high levels of sCD163 and decreased insulin sensitivity in adipose tissue and circulating free fatty acid levels in NAFLD patients [27]. In obese patients with and without high levels of T2DM, sCD163 tended to be associated with decreased insulin sensitivity in adipose and hepatic tissue [26]. Furthermore, sCD163 correlated with hepatic injury and metabolic dysregulation in NAFLD patients and obese individuals before and after lifestyle or surgical intervention [25,28,29]. In addition, sCD163 is linked to IR and is a strong predictor of T2DM [30]. Together, these findings illustrate the utility of sCD163 as a biomarker for MAFLD and underline the interplay between MAFLD macrophage activation and MS.

A high fibrosis stage tended to be significantly associated with clinical progression in our study. This finding is in line with a recent systematic review and meta-analysis that found biopsy-proven fibrosis in NAFLD to be associated with increased liver-related morbidity as well as all-cause mortality [31]. Further, Vilar-Gomez et al. showed that NAFLD patients with cirrhosis were more likely to have liver-related events compared to individuals with NAFLD and F3 fibrosis. Conversely, individuals with F3 fibrosis had a higher cumulated incidence of vascular events and non-hepatic cancers [32]. These findings could explain the low number of liver-related events in our study, as the majority of the included individuals in our cohort had F0–F2 fibrosis.

The present study adds to the relatively sparse data on progression of MAFLD, including serial liver biopsies, and demonstrates that the newly proposed MAFLD inclusion criteria are effective at finding individuals at risk of progression. The small number of individuals not included by the MAFLD criteria are characterized by a less metabolically dysregulated phenotype, and as such could be a subgroup of patients not sufficiently stratified by the NAFLD criteria. To elaborate, Nguyen et al. found that individuals with NAFLD but not MAFLD had a significantly higher rate of advanced fibrosis than those that encompassed both MAFLD and NAFLD criteria, being surpassed only by those who fulfilled the diagnosis of MAFLD but not NAFLD [14]. These findings highlight one weakness of this study. Only patients with an initial diagnosis of NAFLD were enrolled, and as a consequence, we were unable to test the full diagnostic capabilities of MAFLD criteria. Two recent studies have shown that MAFLD patients could be worse off than those fulfilling both MAFLD and NAFLD criteria. Nguyen et al. found that this group had higher rates of advanced fibrosis and higher 15-year cumulative all-cause mortality compared to individuals with only NAFLD or NAFLD and MAFLD [14]. Similarly, Niriella et al. found that non-NAFLD MAFLD patients had a higher risk of cardiovascular events and developing T2DM compared to those fulfilling the NAFLD diagnostic criteria [15].

The retrospective design of our study brings with it the possibility of selection bias with regard to indication for liver biopsy and scoring of liver biopsies, although the latter was controlled for since most biopsies were scored by pathologists for diagnostic purposes. The lack of data regarding the indication for liver biopsy at inclusion as well as follow-up is a limitation to this study as well. Further, the use of liver biopsies might have led to a sampling error at either biopsy, leading to inaccurate characterization of disease activity and/or fibrosis stage [33]. Approximately half of the patients were treated with lifestyle intervention at inclusion. Data on the exact type of lifestyle intervention and data from follow-up visits to determine the biochemical and clinical effect of the intervention were not available. For these reasons, we were not able to assess whether lifestyle intervention played a role in the progression of disease. Finally, the cohort size of 129 patients, with 31 having a follow-up biopsy, limits the robustness of data, warranting new studies with longer follow-up periods and larger cohorts. Multicenter prospective studies as well as nationwide retrospective epidemiological studies to assess short- and long-term risk factors for disease progression are warranted. These data are especially important for the design of future clinical trials investigating interventions in MAFLD as correct risk stratification and disease monitoring in such trials remain challenging. 

In summary, this study demonstrates that MAFLD displays a variable natural history. Our study adds to recent papers demonstrating progression to steatohepatitis and fibrosis from simple steatosis, especially in individuals with more severe metabolic dysregulation. However, clinical progression was rare, and individuals with severe histological disease were at higher risk of progression. sCD163 was an independent predictor of MAFLD with steatohepatitis and high sCD163 levels was a risk factor for clinical progression along with female sex, low creatinine, a high NAS score, a high activity score and a high ballooning score on the diagnostic biopsy. Furthermore, we demonstrate the utility of the novel MAFLD criteria in individuals with biopsy-verified NAFLD.

## Figures and Tables

**Figure 1 jcm-11-02271-f001:**
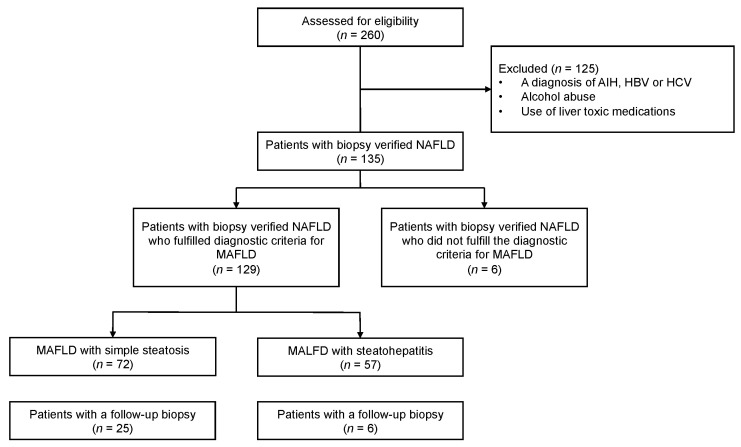
Flowchart for inclusion and separation in categories based on the presence of MAFLD. The number of individuals with follow-up biopsies are shown.

**Table 1 jcm-11-02271-t001:** Baseline characteristics. Discrimination of simple steatosis and steatohepatitis based on SAF score ≥ 5. Data presented as median values and interquartile ranges are in parentheses.

	MAFLD (*n* = 129)	MAFLD Simple Steatosis (*n* = 72)	MAFLD Steatohepatitis (*n* = 57)	Non-MAFLD (*n* = 6)
Sex (male/female)	66/63 (51%/49%)	39/33 (54%/46%)	27/30 (53%/47%), *p* = 0.48	3/3 (50%/50%)
Age (years)	46 (31–546)	42 (27–54)	50 (33–59), *p* = 0.11	33 (21–62)
BMI (kg/m^2^)	33 (29–36)	31 (28–35)	**34 (29–37), *p* = 0.03**	23 (20–23)
Diabetes (*n*)	30 (23%)	7 (10%)	**23 (40%), *p* < 0.0001**	1 (17%)
Metabolic syndrome (*n*)	89 (69%)	44 (61%)	**45 (79%), *p* = 0.03**	1 (17%)
Weekly alcohol consumption (0–7/8–14/15–21/>21) ^a^	111/9/4/2	63/5/2/0	48/4/2/2, *p* = 0.46	5/1/0/0
MELD score	6 (6–7)	6 (6–7)	6 (6–7), *p* = 0.48	6.5 (6–8)
Histology:				
Activity (*n* = A0/1/2/3/4)	30/39/31/20/9	30/39/3/0/0	**0/0/28/20/9, *p* < 0.0001**	2/4/0/0/0
Fibrosis (*n* = F0/1/2/3/4)	63/44/13/6/3	57/14/1/0/0	**6/30/12/6/3, *p* < 0.0001**	4/1/1/0/0
Liver stiffness (kPa)	7.4 (5.4–11.4)	5.3 (4.2–7.4)	**9.4 (6.8–12.5), *p* = 0.001**	N/A
Haemoglobin (mmol/L)	9.2 (8.8–9.8)	9.2 (8.8–9.7)	9.2 (8.6–10.0), *p* = 0.82	9.5 (9.5–10.2)
HbA1c (mmol/mol)	40 (35–47)	38 (34–41)	**46 (40–53), *p* < 0.0001**	26 (23–84)
ALT (IU/L)	111 (78–155)	110 (73–157)	120 (82–154), *p* = 0.45	124 (98–151)
Alkaline phosphatase (IU/L)	84 (69–111)	82 (68–113)	85 (71–110), *p* = 0.69	91 (60–123)
Bilirubin (umol/L)	8 (6–11)	8 (6–11)	8 (6 -11), *p* = 0.45	14 (10–21)
Ferritin (ug/L)	233 (122–390)	233 (102–382)	253 (152–452), *p* = 0.35	183 (129–715)
CRP (mg/L)	3 (2.5–6.0)	2.65 (1.45–4.6)	**3.7 (1.7–9.2), *p* = 0.03**	1.25 (0.6–3)
GGT (U/I)	104.5 (69–214)	90 (46–150)	126 (88–224), *p* = 0.07	56 (42–70)
Amylase (U/I)	29 (21- 43)	31 (24- 51)	**24 (17–44), *p* = 0.002**	25 (14–46)
Triglycerides (mmol/L)	1.9 (1.4– 3.0)	1.8 (1.3–2.8)	**2.3 (1.5–3.7), *p* = 0.03**	1.3 (1.0–2.2)
HDL (mmol/L)	1.2 (1.0–1.3)	1.2 (1.0–1.3)	**1.1 (0.9–1.2), *p* = 0.04**	1.5 (1–1.6)
LDL (mmol/L)	2.4 (3–3.8)	3.2 (2.7–4.8)	**2.5 (2.1–3.6), *p* = 0.006**	1.4 (1.2–2)
Leukocytes (10^9^/L)	6.9 (5.6–8.5)	6.7 (5.6–8.2)	7–0 (6.1–9.0), *p* = 0.22	8 (6.1–10.4)
sCD163 (mg/L)	2.7 (2.1–4.2)	2.2 (1.9–2.9)	**4.12 (2.6–5.6), *p* < 0.0001**	2.0 (2.0–2.0)

^a^: weekly units of alcohol (1 unit equals a serving of 12 g of ethanol. Fields with bold lettering marks parameters significantly different in MAFLD with steatohepatitis compared to MAFLD with simple steatosis. The exact *p*-value is provided in the field.

**Table 2 jcm-11-02271-t002:** Odds ratios (OR) with 95% confidence intervals (CI) for univariate baseline factors associated with clinical progression.

	OR (95% CI)	*p*-Value
Sex (male)	0.12 (0.03–0.59)	0.01
Fibrosis stage	1.57 (0.95–2.58)	0.07
Activity score	1.74 (1.08–2.82)	0.02
Ballooning	2.03 (1.01–4.10)	0.04
NAS score	1.74 (1.06–2.55)	0.02
sCD163	2.6 (1.27–5.31)	0.01
Creatinine	0.95 (0.92–0.99)	0.03

## Data Availability

Data are contained within the manuscript and are available upon request to the corresponding author.

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
