# Peer review of "Clinical Progression of Metabolic-Associated Fatty Liver Disease Is Rare in a Danish Tertiary Liver Center"

_jcm, 2022, doi:10.3390/jcm11092271_

Round 1

Reviewer 1 Report

They found that their own clinical progression was associated with female sex, high creatinine, high sCD163, and disease severity in the diagnostic liver biopsy. Steatohepatitis was associated with metabolic and inflammatory parameters including fibroscan. Disease progression was seen in only 11% mainly related to more severe histological disease at baseline. I believe these findings would contribute to future preventive therapy to reduce progressive steatohepatitis.

  1. What was the definition of “simple steatosis” and “steatosis” in liver biopsy?
  2. I would like to know severity of liver disease such as Child score or MELD score. How about that?
  3. In patient characteristic how often did they drink alcohol?
  4. Which scoring system or markers were best to predict progressive steatosis?

Author Response

Reviewer #1:

They found that their own clinical progression was associated with female sex, high creatinine, high sCD163, and disease severity in the diagnostic liver biopsy. Steatohepatitis was associated with metabolic and inflammatory parameters including fibroscan. Disease progression was seen in only 11% mainly related to more severe histological disease at baseline. I believe these findings would contribute to future preventive therapy to reduce progressive steatohepatitis.

Answer: Thank you for your comments, we agree and hope that the results may contribute to future improvements in MAFLD.

1: What was the definition of “simple steatosis” and “steatosis” in liver biopsy?

Answer: MAFLD with simple steatosis was defined as a NAS of <5 or using the SAF-algorithm (page 3, lines 83-85). Steatosis in the biopsies were graded from 1-3. Grade 1 (3-33% steatosis), grade 2 (33-66% steatosis) and grade 3 (>66% steatosis). This has been clarified in the revised manuscript (page 3, line 82).

2: I would like to know severity of liver disease such as Child score or MELD score. How about that?

Answer: Thank you for this suggestion. Only 3 patients presented with F4 fibrosis, hence we did not include Child-Pugh class. However, we have added MELD-score in Table 1 (Page 4).

3: In patient characteristic how often did they drink alcohol?

Answer: Weekly alcohol consumption has been added to Table 1 (Page 4) There was no statistical difference in weekly alcohol consumption between the patients with MAFLD with simple steatosis and MAFLD with steatohepatitis.

4: Which scoring system or markers were best to predict progressive steatosis?
Answer: This is an interesting point that we would like to elaborate. Unfortunately, only 3 of the 31 patients with follow-up biopsy had progression of their steatosis during follow-up and thus statistical analysis is not relevant.

Reviewer 2 Report

Well-written paper looking at important topic. 

As already acknowledged in the paper, there a a few limitations in the study design that might make it difficult to interpret the results.

1) Line 145-146, there was no differences in alcohol consumption between the 2 groups. What is the average amount of alcohol consumption in each group? will there be a component of alcoholic steatohepatitis?

2) the median time of followup is still relatively short and probably one would expect there should not be significant progression in liver fibrosis

3) were you able to get any information on the level of activity/exercise, diet, new medications started on co-morbidities (e.g. dyslipidemia) between the groups as all these can affect the results

4) how do you explain why there is statistically difference in Fibroscan score in MAFLD steatohepatitis group but no in Fibrosis from histology?

5) what you do think /postulate the reasons why female sex is associated with progression?

6) there were some patients who regressed, did you look at the factors that predict regression?

Author Response

Reviewer #2:

Well-written paper looking at important topic. As already acknowledged in the paper, there are a few limitations in the study design that might make it difficult to interpret the results.

Answer: Thank you for your comments and for raising some important points that we have added in the revised manuscript.

1: Line 145-146, there was no differences in alcohol consumption between the 2 groups. What is the average amount of alcohol consumption in each group? will there be a component of alcoholic steatohepatitis?
Answer: Weekly alcohol consumption has been added to Table 1 (Page 4). In general, the alcohol consumption was low in all groups and there was no statistical difference in weekly alcohol consumption between the patients with MAFLD with and without steatohepatitis. Thus, we do not consider the steatohepatitis to be alcohol-related.

2: The median time of followup is still relatively short and probably one would expect there should not be significant progression in liver fibrosis
Answer: We agree that it could be a problem with short follow-up time. In our study, the follow-up had a wide range of 172 days – 25.5 years.  Recent data indicate that approximately 20% are rapid progressors (Ref: Singh S, Allen AM, Wang Z, Prokop LJ, Murad MH, Loomba R. Fibrosis progression in nonalcoholic fatty liver vs nonalcoholic steatohepatitis: a systematic review and meta-analysis of paired-biopsy studies. Clin Gastroenterol Hepatol, 13(4), 643-654.e641-649; quiz e639-640 (2015)). However, additional studies with long follow-up and larger cohorts are warranted. This consideration has been added in the revised manuscript (page 8, lines 324-326).

3: Were you able to get any information on the level of activity/exercise, diet, new medications started on co-morbidities (e.g. dyslipidemia) between the groups as all these can affect the results
Answer: Unfortunately, no data on the level of activity/exercise were available. Sixty patients were encouraged to initial lifestyle intervention. Significantly more patients in the MAFLD with steatohepatitis group than in the simple steatosis group were treated with lifestyle intervention at diagnosis. This consideration has been added in the revised manuscript (page 5, lines 145-148). The number of patients starting new treatment during follow-up is mentioned in page 5, lines 148-149).  An elaborating paragraph on patients starting new treatment during follow-up was added (page 6, lines  200-202).

4: How do you explain why there is statistically difference in Fibroscan score in MAFLD steatohepatitis group but no in Fibrosis from histology?
Answer: We are sorry that this has not been clear. There are in fact statistically significant differences between MAFLD with steatohepatitis compared to MAFLD with simple steatosis with regard to activity and fibrosis. This information has been added to Table 1 as an asterisk, and in page 5 line 155.  

5: What you do think /postulate the reasons why female sex is associated with progression?
Answer: In our cohort, the females were significantly older than the males, which may explain some or even all of the effect. Generally, female sex is normally considered a protective factor with regards to fibrosis development.

6: There were some patients who regressed, did you look at the factors that predict regression?
Answer: This is an interesting point, but as there were only two of 6 patients regressing from steatohepatitis, and 4 of 31 patients with regression in fibrosis stage, meaningful statistical analysis is not possible.

Reviewer 3 Report

In this manuscript, Tea Lund Laursen et al studied a MAFLD cohort at a Danish tertiary liver center in order to understand the potential risk of disease progression in patients with MAFLD.  The authors retrospectively assessed 129 patients with biopsy-proven MAFLD with 31 of them had a follow-up biopsy after a median of 287 days.  Their results show that 20 patients had the same fibrosis stage on the follow-up biopsy, 7 progressed, and 4 regressed.  14 patients progressed clinically with the median follow-up time 3.8 years.  Several factors were also found to be associated with clinical progression.  This study seems interesting, however, this study seems interesting, however, the cohort size is small.

Major comments:

  1.  This study only included 129 patients with 31of them were followed up.  The cohort size is small.
  2. The authors did not consider any medical interventions in the study.  Since patients were already diagnosed with MAFLD, different medical interventions between simple steatosis patients and steatohepatitis patients may cause the difference in disease progression.

Author Response

Reviewer #3:
In this manuscript, Tea Lund Laursen et al studied a MAFLD cohort at a Danish tertiary liver center in order to understand the potential risk of disease progression in patients with MAFLD.  The authors retrospectively assessed 129 patients with biopsy-proven MAFLD with 31 of them had a follow-up biopsy after a median of 287 days.  Their results show that 20 patients had the same fibrosis stage on the follow-up biopsy, 7 progressed, and 4 regressed.  14 patients progressed clinically with the median follow-up time 3.8 years.  Several factors were also found to be associated with clinical progression.  This study seems interesting, however, this study seems interesting, however, the cohort size is small.

Answer: Thank you for your comments, which we have answered below.

Major comments:

1:  This study only included 129 patients with 31 of them were followed up.  The cohort size is small.

Answer: This is a retrospective study, where we assessed all consecutive patients with a liver biopsy and a diagnosis of NAFLD: We agree that it could be interesting to assess a larger cohort in the future. However, this study was designed as part of a quality database in our department and provide interesting knowledge that hopefully pave the road for further studies on the subject.

2: The authors did not consider any medical interventions in the study.  Since patients were already diagnosed with MAFLD, different medical interventions between simple steatosis patients and steatohepatitis patients may cause the difference in disease progression.

Answer: Twenty-three patients were initially treated with Vitamin E supplementation and further 16 (see page 6, lines 200-202) started during follow-up. In our statistical analysis Vitamin E supplementation had no effect on the risk of clinical or histological progression. Three patients received GLP-1 at baseline, and further 7 were initiated during follow-up (page 6, line 201). This number of patients is too small for meaningful statistical interpretation of data.

Reviewer 4 Report

In this manuscript by Laursen et al, the authors investigated retrospectively a cohort of biopsy-verified NAFLD with the new criteria for metabolic associated fatty liver disease (MAFLD) and analyzed the clinical parameters in terms of disease severity. They found that patients with steatohepatitis had higher BMI, liver stiffness, sCD163 levels among several other baseline characteristics. Clinical progression was relatively rare and occurred in more histologically severe cases. This is a well-constructed and written study. However, a number of minor concerns need to be addressed. Specific comments are as follows:

1. The term in the title "Progression of metabolic associated fatty liver disease is rare" may be misleading as this refers to the proportion of patients clinically diagnosed with more severe conditions during the course of study. At the beginning, 44% already had steatohepatitis (which was progressed from simple steatosis). Additionally, progression determined by histology had a higher rate.

2. Is Figure 2 necessary as the pie chart does not add more information than what the text reveals.

3. It will be interesting to discuss the observation that LDL is lower in more severe MAFLD.

4. It is suggested to have a separate paragraph discussing the limitation of the study.

Author Response

Reviewer #4:

In this manuscript by Laursen et al, the authors investigated retrospectively a cohort of biopsy-verified NAFLD with the new criteria for metabolic associated fatty liver disease (MAFLD) and analyzed the clinical parameters in terms of disease severity. They found that patients with steatohepatitis had higher BMI, liver stiffness, sCD163 levels among several other baseline characteristics. Clinical progression was relatively rare and occurred in more histologically severe cases. This is a well-constructed and written study. However, a number of minor concerns need to be addressed. Specific comments are as follows:

Answer: Thank you for your positive feedback and constructive comments, which we have addressed below.

1: The term in the title "Progression of metabolic associated fatty liver disease is rare" may be misleading as this refers to the proportion of patients clinically diagnosed with more severe conditions during the course of study. At the beginning, 44% already had steatohepatitis (which was progressed from simple steatosis). Additionally, progression determined by histology had a higher rate. 
Answer: Thank you for addressing this, we have changed the title to “Clinical progression of metabolic associated fatty liver disease is rare in a Danish tertiary liver center”.
2
: Is Figure 2 necessary as the pie chart does not add more information than what the text reveals.
Answer: We agree, Figure 2 has been removed.
3
: It will be interesting to discuss the observation that LDL is lower in more severe MAFLD.
Answer: This is an interesting observation. One hypothesis could be that a larger proportion of patients in the MAFLD with steatohepatitis were treated with statins prior to their fatty liver diagnosis compared to MAFLD patients with simple steatosis. Unfortunately, data on statin use was not available in our data set.
4
: It is suggested to have a separate paragraph discussing the limitation of the study.
Answer: We agree that it is important to mention the limitations of the study, we have included this in the last part of the discussion (page 8, lines 310-312 and lines 320-326)

Round 2

Reviewer 2 Report

Thank you for the clarifications and revisions.

Author Response

Thank you very much for your comments.

Reviewer 3 Report

The authors addressed my questions in the revision . I agree it for publication.

Author Response

Thank you very much for your comments.